# Effect of a Cross-Training and Resistance Exercise Routine on IL-15 in Adults with Type B Acute Lymphoblastic Leukemia during the Induction Phase: Randomized Pilot Study

**DOI:** 10.3390/jfmk9010004

**Published:** 2023-12-21

**Authors:** Adán Germán Gallardo Rodríguez, Irma Olarte Carrillo, Adolfo Martínez Tovar, Rafael Cerón Maldonado, Emmanuel Martínez Moreno, Christian Omar Ramos Peñafiel

**Affiliations:** 1Medicine Research Doctoral Program, Escuela Superior de Medicina, Instituto Politécnico Nacional, w/n Plan de San Luis y Díaz Mirón, Casco Santo Tomás, Miguel Hidalgo, Mexico City 11340, Mexico; nutriologo.agallardo8@gmail.com; 2Hematology Department, Hospital General de México “Dr. Eduardo Liceaga”, 148 Dr. Balmis, Doctores, Cuauthémoc, Mexico City 06720, Mexico; drmanumartz@gmail.com; 3Molecular Biology Department, Hematology Department, Hospital General de México “Dr. Eduardo Liceaga”, 148 Dr. Balmis, Doctores, Cuauthémoc, Mexico City 06720, Mexico; irmaolartec@gmail.com (I.O.C.); mtadolfo73@hotmail.com (A.M.T.); cmrafael.bh@gmail.com (R.C.M.)

**Keywords:** Interleukin-15, exercise, acute lymphoblastic leukemia, chemotherapy, body composition

## Abstract

IL-15 is a proinflammatory myokine essential for activating NK cells and CD8+ T lymphocytes, and its overexpression has been related to reducing overall survivorship in patients with acute lymphoblastic leukemia (ALL). Physical exercise has been shown to be safe, feasible, and beneficial in hematological cancers. Exercise requires the activation of muscles that secrete cytokines, such as IL-15, causing immune mobilization. The objective was to compare the outcomes of two training routines on IL-15 and survival prognosis in adult patients diagnosed with ALL. A blind randomized clinical study was carried out where twenty-three peripheral blood samples were obtained pre and postexercise intervention from patients categorized into three types of intervention: the resistance exercise group (REG), the cross-training exercise group (CEG), and the control group (CG). Changes in IL-15 levels during the intervention were not significant in any of the groups (CG *p* = 0.237, REG *p* = 0.866, and CEG *p* = 0.678). However, 87.5% of patients who received an exercise intervention achieved remission, while only 21.73% experienced a relapse. There were no deaths during the study. Although IL-15 level adaptation in the REG and the CG performed similarly, the REG induced a better clinical outcome. Resistance exercises may help improve survival prognosis and reduce relapses in patients with ALL.

## 1. Introduction

Interleukin-15 (IL-15) is a 14–15 kDa proinflammatory cytokine. It acts upon a heterotrimeric receptor (IL-2/IL-15R β, γc and IL-2/IL-15R β, γc) regulating the development, proliferation, survival, and differentiation of immune cells, especially T, B, and Natural Killer cells (NK) [1], causing a memory effect on CD8+ T lymphocytes and sharing an effect with IL-2 [2], only differing in effect on the stimulation of immuno-suppressive T-regulatory cells [3]. IL-15 is essential for the activation of NK cells and CD8+ T lymphocytes; its activation uses a common cytokine receptor (CD132), which coactivates other cytokines (IL-2, IL-4, IL-7, IL-9, and IL-21), a receptor called β that activates IL2/IL-15, and a third one which is activated through the Janus kinase pathway; its overactivation has been related to both autoimmune diseases and lymphoid neoplasms [4]. IL-15 is secreted as a mature protein by leucocytes (macrophages and monocytes), dendritic cells, stromal cells, and muscle, in response to a physical stimulus, affecting adipose tissue and muscle fibers [5]. The antiapoptotic properties of IL-15 have been previously described, causing a possible selective advantage of neoplastic cell growth by promoting tumor growth, metastasis, and infiltration of malignant cells in other target organs [6]. In hematological cancers, IL-15 elevation has been identified in T-granular lymphoma, as its expression is associated with a greater risk of infiltration into the central nervous system [4,7]. Acute lymphoblastic leukemia is a neoplasm characterized by the uncontrolled proliferation of a lymphoid progenitor (of the B or T phenotype) that invades the bone marrow or other extramedullary tissues, displacing normal hematopoiesis [8]. In adults with acute lymphoblastic leukemia (ALL), IL-15 overexpression has been associated with decreased survival [9], possibly due to the direct effect of soluble and receptor-bound IL-15 on tumor growth and immune escape [10]. It has been described that leukemic cells in relapsed ALL patients produce a broad spectrum of cytokines, such as IL-15, causing blast growth [11]. Muscle mass depletion is another factor related to the survival prediction of patients with ALL. Previous studies have shown that low muscle mass is expected regardless of cancer stage and is an independent predictor of poor physical function, lower quality of life, surgical complications, cancer progression, and reduced survival [12,13,14]. Therefore, exercise has become a relevant therapy in the treatment of ALL. Physical exercise training has been demonstrated to be safe, feasible, and beneficial in different cancer malignancies, including hematological cancers, due to its effects on body composition, quality of life, sensitivity to chemotherapy, and decreased adverse effects [15,16]. Exercise requires the activation of muscles that can secrete molecules and peptides during contraction, myogenesis, and muscle remodeling, in addition to cytokines such as IL-15, which results in immune mobilization and the accumulation of CD8 T cells, accountable for antitumor effects [17,18]. Therefore, the objective of this protocol was to compare the outcomes of two training routines on IL-15 and survival prognosis in adult patients recently diagnosed with de novo ALL.

## 2. Materials and Methods

Twenty-three patients with a recent diagnosis of ALL and who began their induction therapy (CALGB 10403 treatment scheme) were included in the study, from which one sample was obtained before and after the intervention (46 total samples) between May 2021 and September 2022 at the Hematology Services of Hospital General de México “Dr. Eduardo Liceaga”. Samples were taken prior to initiating chemotherapy and after the first chemotherapy regimen (28 days).

The exclusion criteria included adult patients with the following characteristics: (1) with any bleeding and/or infection during hospitalization; (2) who were immobile or unable to perform physical activity; (3) with the presence of any central nervous system diseases that make movement impossible; (4) with changes in cardiac performance; (5) with bone marrow or central nervous system relapse; and (6) who were referred from another hospital and who were treated in our service. All patients gave their written informed consent for the protocol and the data gathering. According to the Declaration of Helsinki, this three-arm, prospective, randomized, blinded, open-label pilot study (ratio 1:1:1) was conducted. A simple randomization was done using EpiInfo (v.7.2) for this study.

This study belongs to a more extensive protocol approved by the Biosafety, Ethics, and Research Committee of the Hospital General de México “Dr. Eduardo Liceaga” with the protocol number (HGMDI/21/204/03/46) and registered at ClinicalTrials.gov with the registration code NCT05059847.

### 2.1. Anthropometry and Body Composition

The BMI, body weight, and height of the patients were measured with a mechanical scale with a calibrated stadiometer of the BAME brand (Mexico), with a weighing capacity of up to 160 kg, and a scale to measure height up to 1.95 m. The body composition was measured with an 8-electrode bioimpedance meter (SECA mBCA 525). The patient was placed in a supine position with their arms and legs slightly separated from the body; with electrodes connected to each hand: one at the radiocarpal joint and the other at the level of the carpometacarpal joint; on the feet, the electrodes were located at the tibial–tarsal joint and the other at the level of the tarsal bones. These electrodes were attached to a belt placed on the patient’s legs. The bioimpedance meter allowed the measuring of different body compartments like free fat mass (FFM), percentage of free fat mass (PFFM), body fat mass (BFM), percentage of body fat mass (PBFM), skeletal muscle mass (SMM), and percentage of skeletal muscle mass (PSMM).

### 2.2. Establishing IL-15

IL-15 levels were determined with ELISA in patients with leukemia. All washes between steps were done with 1× PBS, pH = 7.4 (Gibco, Life Technologies, Grand Island, NY, USA). Five milliliters of peripheral blood from patients with leukemia were obtained in a yellow cap Vacutainer tube (Vacutainer tubes, BD Diagnostics Franklin Lakes, New Jersey, NJ, USA) and centrifuged at 2500 rpm for 5 min to separate the serum; it was stored at −80 °C until used. A total of 100 µL of the serum sample was incubated in duplicate in sterile 96-well plates (Corning, New York, NY, USA) and incubated overnight at 4 °C. Subsequently, it was blocked with 1% albumin solution (A-7030, Sigma®-Aldrich, St. Louis, MO, USA) for 2 h at 37 °C. It was washed and incubated with Anti-IL-15 (MA5-23729, Thermo Fisher Scientific, Waltham, MA, USA) for 2 h at 37 °C. After washing, it was incubated with the antibody goat anti-mouse IgG-HRP (sc-2031, Santa Cruz Biotechnology, Dallas, TX, USA) for 2 h at 37 °C for later immunodetection with TMB Peroxidase EIA Substrate Kit Solution (Bio-Rad Laboratories, Inc., Hercules, CA, USA), in the iMARK equipment (Bio-Rad Laboratories, Inc.), the corresponding reading of the ELISA plates was performed at a wavelength of 655 nm.

### 2.3. Exercise Intervention

Patients were randomly allocated to one of the following groups: the control group (CG), the cross-training exercise group (CEG), and the resistance exercise group (REG). Exercise routines were performed during hospitalization through three weekly exercise sessions lasting 50 min each. The exercise routines were managed and registered by an expert trainer. The resistance exercise protocols were based on recent protocols and training guidelines for hematological cancer patients [16,19,20,21].

Patients in the CEG performed directed individualized routine exercises focused on stability, abdominal and overall body strength, and joint mobility and stability. Patients began with a five-minute warm-up that included dynamic and isometric exercises to increase body temperature. The main phase corresponded to a 40 min workout of seven different exercises and 5 minutes of stretching exercises at the end of the routine. A broomstick and body weight were used to complement the routine performance. Intensity was settled by an RPE of 3–6 (similar to 50–75% of heart rate reserve). The improvements in the exercise routines were structured according to the progression and condition of the patient, carrying out three to five sets of eight to fifteen repetitions each.

Patients in the REG performed individual and directed resistance exercises, including body-weight exercises and light-weight dumbbell exercises (one to five kg). The planned exercise routine was based on ten different activities. Intensity, duration, and frequency were adjusted according to the initial test measurements. The CG received general activity recommendations (walking daily for at least 30 min) to prevent sedentarism during hospitalization, but no structured physical activity was followed. 

Routinely, to assess the well-being of patients, complete blood cell counts and vital signs were evaluated by the service’s nursing team. Patients skipped exercise sessions in any of the following situations: platelets count <20 × 10^3^/µcL, hemoglobin <6.0 g/dL, temperature >38 °C, and adverse effects post chemotherapy including any case of bleeding.

### 2.4. Statistical Analysis

Measures of central tendency were used to describe the characteristics of the patients in each group. The ANOVA test was used to compare independent and related samples of body composition outcomes. The Mann–Whitney U test was performed to calculate independent and related samples of blood count levels and IL-15 observance outcomes in patients with ALL. In addition, Kaplan–Meier estimates were made to analyze the probability of relapse or failure to the induction treatment. Regardless of the median value, differences between groups were analyzed with a Log-Rank test. Data are expressed in median values. A *p* < 0.05 was established as a statistical difference. Statistical analysis was executed using the statistical software SPSS version 25, and figures were generated with GraphPad Prisma version 7 software.

## 3. Results

A total of 33 patients were included in the protocol; however, only 23 baseline and postintervention samples [CG (*n* = 7), CEG (*n* = 9), and REG (*n* = 7)] could be collected and analyzed. The rest of the samples (*n* = 10) were not processed due to hemolysis. The group demographic, anthropometric, body composition, and biochemical characteristics were similar. The mean age of the groups was 32.14 (±9.83) years old for CG, 26.86 (±6.17) years old for REG, and 23.33 (±6.30) years old for CEG with a *p* = 0.086. At baseline, none of the body composition or biochemical outcomes showed a statistical difference between groups; meanwhile, IL-15 observance values exhibit statistical differences between groups (Table 1).

At the end of the intervention, no differences were shown in the CG compared to the baseline values. However, in the REG, significant differences were found in body composition, characterized by a substantial loss of FFM, PFFM, and SMM (*p* = 0.011, 0.005, and 0.003, respectively) and an increase in BFM and PBFM (*p* = 0.012 and *p* = 0.005). In CEG, the only variables that showed differences when the follow-up concluded were the BMI (*p* = 0.042) and the FFM (*p* = 0.029), which decreased significantly. No differences were found when comparing outcomes from the three groups (Table 2).

On the other hand, the biochemical variables of hemoglobin (*p* = 0.466), neutrophils (*p* = 0.826), leukocytes (*p* = 0.263), and platelets (*p* = 0.097) did not show significant differences between the groups at the end of the intervention. Within the groups, a statistical difference was only located in leucocyte levels (*p* = 0.028) in the REG and hemoglobin (*p* = 0.018) in the CEG. The results of the biochemical variables are shown in Table 3.

This study quantified observance (650 nm) in 23 pre and 23 postinduction phase samples from adults with ALL. The observance of IL-15 varied considerably depending on the group to which each patient was assigned. Samples were taken from healthy subjects (*n* = 5) as a standardization method. The difference between baseline medians and healthy subjects was calculated using Fisher’s exact test, and, for each group, it showed a statistically significant difference (CG *p* = 0.015, REG *p* = 0.015, and CEG *p* = 0.021). 

The median observance at the end of the intervention for the CG was 0.036, for the REG was 0.042, and for the CEG was 0.033 (*p* < 0.032), where a statistical difference was localized between intervention groups. The difference in medians of each of the pre and postintervention groups was measured using the Wilcoxon test; however, the changes during the intervention did not turn out to be significant (CG *p* = 0.237, REG *p* = 0.866, and CEG *p* = 0.678) (Figure 1).

### 3.1. Correlation of Physical Training with Clinical Outcomes in Adult Patients with ALL

Of the 23 patients included, 18 (78.26%) achieved remission at the end of induction. In contrast, only five (21.73%) presented a relapse at the end of induction (three in the central nervous system and two in the bone marrow). None of the patients died during the study, so it was decided to follow them for up to 500 days to confirm their survival. At the end of the 500 days, only six (26.08%) died. The leading causes of death were disease recurrence after complete remission (two), chemotherapy-related complications, and disease progression (four). In addition, the median survival days were 412.71 for the CG, 445.57 for the REG, and 488.5 for the CEG. Forty-five days after the start of the induction phase, a Measurable Residual Disease (MRD) study was performed to corroborate the residual blasts in the bone marrow after chemotherapy. A total of 21 (91.30%) patients achieved a positive MRD (>0.01) (Table 4).

### 3.2. Correlation of IL-15 with the Clinical Prognosis of Patients

Of the 23 patients included, only 6 (26.08%) died during follow-up; however, none presented an overquantification of IL-15 according to the new parameters, so it was not statistically significant (*p* = 0.198). IL-15 values of <0.036 were a protective factor associated with mortality (OR = 0.727; 95% CI 0.563–0.939; and r2 = −0.273). In response to the induction scheme and relapse after treatment, only one (16.7%) patient out of six presented an overquantification of IL-15. However, this factor was not clinically or statistically significant in either outcome (OR = 1.087; 95% CI 0.160–7.391; and r2 = −0.016; *p* = 0.715).

Finally, the interaction of the primary dependent variables (intervention, exercise, and IL-15 overquantification groups) on survival was analyzed using a Kaplan–Meier test. When separated by intervention (Figure 2a) and observance of IL-15 cut-off points (Figure 2c) the Log-Rank test was not significant (0.086 and 0.112, respectively); however, it was observed that when comparing the patients who exercised against the control group (Figure 2b), statistical significance was found, proving the benefits of physical intervention in the adult population with ALL (0.027).

## 4. Discussion

This protocol evaluated the impact of two training routines on the regulation of IL-15 and the effect on the overall survivorship of patients with acute lymphoblastic leukemia during the first month of treatment (induction). Initially, baseline IL-15 observance levels appeared augmented in ALL patients compared to healthy individuals. These results were consistent with the review by Sindaco et al., where it is described that, in patients with ALL, IL-15 is overexpressed. This increase was associated with disease severity and a greater risk of infiltration to the central nervous system due to PSGL-1 and CXCR3 regulation which attract malignant cells to the central nervous system [22]. Contrary to what was found in our population, Wadai found a significant depletion in IL-15 levels in an Iraqi population of 21 patients with ALL compared to the healthy population [23]. When analyzing the results per group, the postintervention IL-15 observance values decreased in the CG and the REG compared to baseline levels. However, the values were not statistically significant (*p* = 0.237 and *p* = 0.866, respectively). On the other hand, in the CEG, the effect was different since, at the end of the intervention, the levels of IL-15 compliance increased, although they were not significant (*p* = 0.678). The behavior of IL-15 in the CG patients can be explained by immune reconstitution since, being patients who receive cytotoxic chemotherapy, the lymphoid compartments can be modified, including CD4+, CD8+ cells, and NK cells (CD56+), which can modify the expression of cytokines [24]. In a preliminary report on the recovery of immune system parameters in children with ALL after chemotherapy, they described that the number of NK cells after completion of treatment did not differ from the expected values in healthy children [25], which would translate into a decrease in the amount of NK and, therefore, an indirect reduction in IL-15. The decline in IL-15 parameters in the REG after one month of follow-up can be explained by negative feedback; as the body recovers from exercise, IL-15 levels decrease to avoid a chronic inflammatory response. The autocrine effect of IL-15 may mediate this since it has been observed that soluble IL-15 receptors linked to their alpha receptor (sIL-15Rα) can bind to the soluble IL-15 released by the myofibrils of the muscles, thus decreasing their biological activity. These act as primers for IL-15, preventing them from attaching to surface receptors [26]. Evidence has shown an increase in IL-15 postexercise in ALL patients; while in chronic exercise (after three months), there is a decrease in plasma [27]. According to the results obtained in the CEG, it is believed that, as it was a more metabolically demanding routine, due to the type of exercises performed and the use of larger muscle groups, they kept the body in a more significant proinflammatory state compared to the CG and the REG. Therefore, patients may have a training schedule of more than one month to obtain the same results as in the REG. This inconsistency in the IL-15 response to cross-training may result from secretion and regulation mechanisms not yet fully understood. This result coincides with Micielska et al., who reported that IL-15 levels decreased at the baseline test after a high-intensity circuit exercise session (HIT) but increased after five weeks of training. This phenomenon was associated with the levels of contractility of the muscles and the constant exposure of the body to an intense stimulus such as the HIT and the increase in physical performance; however, in our study, it did not have a significant correlation [28].

Another widely described benefit of IL-15 on the body is its pleiotropic role in the lipid and glucose metabolism [29,30]. However, within our study, no significant correlations were found between serum IL-15 levels and absolute fat mass or fat mass percentage. Christiansen et al. described a close relationship between the decrease in IL-15 post training intervention and the loss of fat mass in obese patients [31]; however, in our population, fat mass increased in patients. This effect could be due to the inclusion of glucocorticoids in chemotherapy, as it is one of the main components of the CALGB regimen. The dose and frequency of the glucocorticoids could overcome the oxidative stimulus of IL-15 during their hospital stay [32].

In murine models, it has been revealed that the overexpression of IL-15 and its anabolic/antiatrophic effect is associated with a decrease in skeletal muscle proteolysis and myocyte apoptosis through the suppression of DNA fragments in the pathway of the tumor necrosis factor alpha signaling [33]. This effect allows IL-15 secreted by muscle to take a leading role in muscle hypertrophy. However, our study found no positive associations between IL-15 overexpression and FFM, PFFM, and SMM. These results agree with those of Pérez-López et al., who found no relation between the increase in serum IL-15 and muscle protein synthesis [26]. It should be noted that a characteristic of our population is the sarcopenia associated with fatigue, extended hospital stays, and prolonged bed rest, so it is considered that one hour of any exercise in patients with ALL is not enough to counteract the loss of muscle mass in situations of high energy demand.

The repercussions of physical activity on the different treatment outcomes, such as survival or the presence of measurable residual disease, were measured. During follow-up, patients who did not perform physical activity showed lower survival than the intervention groups. This is unsurprising since various in vivo models, such as clinical trials, have identified the importance of exercise as an adjuvant in cancer treatment [34,35].

The most relevant finding was that individuals with higher levels of IL-15 (>0.036) showed better behavior than those with a lower value. These data suggest that exercise can modify the initial poor prognosis of IL-15 and that a consistent exercise routine, whether resistance or cross-training, promotes a positive effect on the disease through the increase in IL-15 derived from the muscle, potentiating an impact of the immune system on leukemia. Our data coincide with what was proposed by Kurz et al., where performing an aerobic routine increases intratumoral infiltration of IL15Rα+ lymphocytes, reducing tumor growth through the effect of the immune system on the tumor; in addition, adding an IL-15 super-agonist (NIZ985) improves the response to chemotherapy treatment or anti-PD118 drugs. New experimental platforms use the combination of different cytokines, such as IL-12, IL-15, and IL-18, to enhance the effect of the NK cells on leukemic cells [36] or through reprogramming these cells through the CAR-NK cell platform [37]. The role of IL-15 in energy metabolism, muscle growth, and its relationship with cancer and cellular components can be found in Appendix A.

The main limitation of this study was the small sample size, which was related to the fact that it was carried out during the global COVID-19 pandemic, so recruitment and follow-up were affected by considering our hospital a center of reference for all patients nationwide. Another limitation was the ELISA kit used to measure IL-15 in serum since it was not specific for a variant of IL-15 (sIL-15 or IL-15Rα), so it cannot be sure which variant was the one that had a more significant effect or how they behaved during the intervention. It is worth mentioning that the interaction of various cellular and humoral factors mediates the inflammatory response. Considering this, it is necessary to open new lines of research about the interaction of cytokines on the inflammatory reaction of adult patients with ALL, the effects on prognosis and survival, and the relationship with IL-15-mediated NK cells.

## 5. Conclusions

In conclusion, IL-15 is a cytokine altered in adult patients with ALL compared to healthy people. Although the observance levels of IL-15 in the REG and the CG behaved as expected; patients who practiced some training had a better prognosis and survival. The cross-training exercise was a strategy that, although it showed similar results to the REG, turned out to be energetically more demanding for the patients due to the characteristics of the training. It is recommended to include, within the treatment of ALL in adults de novo, a routine of resistance exercises that allows them to reduce the muscular deterioration of the disease and the chronic inflammatory state and improve their prognosis of survival and postinduction relapse.

## Figures and Tables

**Figure 1 jfmk-09-00004-f001:**
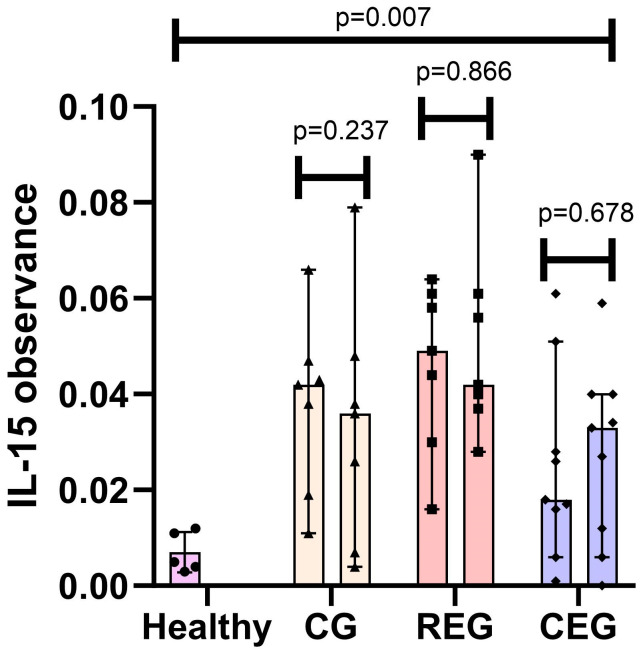
IL-15 observance pre- vs post-intervention classified by group and compared to healthy subjects. • Healthy IL-15 samples; ▲ Control Group IL-15 samples; ▪ Resistance Exercise Group IL-15 samples; ◆ Cross-training Exercise Group IL-15 samples.

**Figure 2 jfmk-09-00004-f002:**
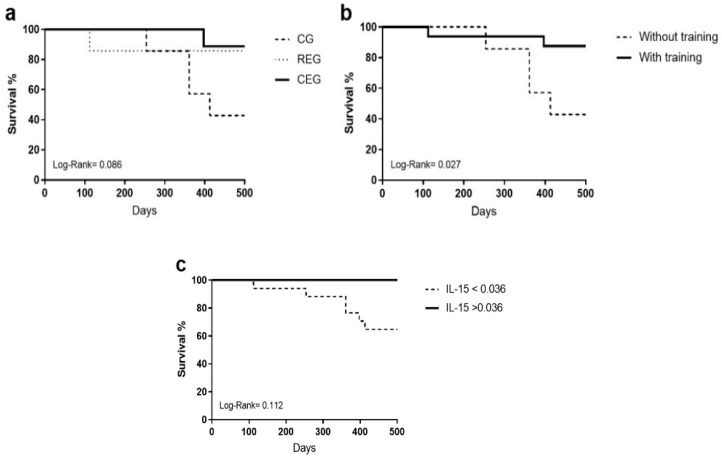
Kaplan–Meier test of the primary dependent variables on the overall survival of adult patients with ALL. (**a**) Assigned intervention, (**b**) with and without training, and (**c**) adjusted cut-off points.

**Table 1 jfmk-09-00004-t001:** Demographic, anthropometric, immunological, and biochemical characteristics of patients at baseline.

	CG (*n* = 7)	REG (*n* = 7)	CEG (*n* = 9)	*p* Value ^a^	*p* Value ^b^
Age (Years)	32.14 ± 9.83	26.86 ± 6.17	23.33 ± 6.30	0.043	0.086
Gender (M:F)	3:4	2:5	1:8		0.352
BMI (kg/m^2^)	28.16 ± 6.29	29.29 ± 4.77	26.52 ± 7.19	0.881	0.680
FFM (kg)	55.17 ± 13.00	63.88 ± 11.53	61.01 ± 14.70	0.244	0.471
PFFM (%)	73.10 ± 17.31	79.72 ± 13.71	83.74 ± 15.59	0.215	0.415
BFM (kg)	22.39 ± 14.76	17.40 ± 11.80	14.22 ± 13.80	0.273	0.500
PBFM (%)	26.90 ± 17.31	20.27 ± 13.71	16.25 ± 15.59	0.215	0.415
SMM (kg)	26.75 ± 7.14	30.78 ± 5.49	29.44 ± 8.10	0.312	0.566
PSMM (%)	35.00 ± 8.08	38.27 ± 6.18	41.59 ± 6.97	0.109	0.208
IL-15 (observances)	0.042 (0.11–0.055)	0.049 (0.016–0.064)	0.018 (0.001–0.061)	0.769	0.007
Hemoglobin (mg/dL)	8.20 (8.9–5.4)	8.00 (6.10–8.70)	6.75 (4.80–12.10)	0.298	0.675
Neutrophiles (×10^3^/µL)	2.30 (0.4–89.4)	0.70 (0.4–160.0)	1.20 (0.40–12.10)	0.490	0.867
Leucocytes (×10^3^/µL)	46.30 (1.3–386)	19.50 (3.10–282.00)	13.45 (1.80–171.50)	0.490	0.675
Platelets (×10^3^/µL)	23.00 (14.0–62.00)	49.00 (8.00–92.00)	35.50 (7.00–109.00)	0.298	0.867

M: Male; F: Female; BMI: Body Mass Index; FFM: Fat-Free Mass; PMLG: Percentage of Fat-Free mass; BFM: Body Fat Mass; PBFM: Percentage of Body Fat Mass; SMM: Skeletal Muscle Mass; PSMM: Percentage Skeletal Muscle Mass; IL: Interleukin. The ANOVA test was used for parametric variables and is expressed in mean (SD). The Mann–Whitney U test was used for non-parametric variables and is expressed in median (range), and the Chi-square test for qualitative variables. ^a^ exercise group vs no exercise group *p* value; ^b^ between groups *p* value. Statistical significance was considered to have a *p*-value of <0.05.

**Table 2 jfmk-09-00004-t002:** Intra and intergroup comparison of changes in body composition.

	CG (*n* = 7)			REG (*n* = 7)			CEG (*n* = 9)			*p* Value ^b^
	Pre	Post	*p* Value ^a^	Pre	Post	*p* Value ^a^	Pre	Post	*p* Value ^a^	
BMI (kg/m^2^)	28.16 ± 6.29	27.17 ± 2.02	0.216	29.29 ± 4.77	26.91 ± 2.17	0.040	26.52 ± 7.19	24.95 ± 1.77	0.042	0.655
FFM (kg)	55.17 ± 13.00	53.66 ± 4.87	0.544	63.88 ± 11.53	55.30 ± 5.26	0.011	61.01 ± 14.70	55.86 ± 4.29	0.029	0.942
PFFM (%)	73.10 ± 17.31	71.41 ± 5.51	0.523	79.72 ± 13.71	74.98 ± 5.95	0.005	83.74 ± 15.59	81.10 ± 4.86	0.070	0.604
BFM (kg)	22.39 ± 14.76	20.67 ± 4.50	0.533	17.40 ± 11.80	19.49 ± 4.86	0.012	14.22 ± 13.80	14.95 ± 3.97	0.445	0.422
PBFM (%)	26.90 ± 17.31	28.54 ± 5.51	0.543	20.27 ± 13.71	25.01 ± 5.95	0.005	16.25 ± 15.59	18.90 ± 4.86	0.070	0.424
SMM (kg)	26.75 ± 7.14	25.45 ± 2.41	0.282	30.78 ± 5.49	27.06 ± 2.61	0.003	29.44 ± 8.10	27.06 ± 2.13	0.072	0.861
PSMM (%)	35.00 ± 8.08	33.95 ± 2.65	0.303	38.27 ± 6.18	36.99 ± 2.86	0.088	41.59 ± 6.97	39.16 ± 2.33	0.265	0.357

BMI: Body Mass Index; FFM: Fat-Free Mass; PFFM: Percentage of Fat-Free Mass; BFM: Body Fat Mass; PBFM: Percentage of Body Fat Mass; SMM: Skeletal Muscle Mass; PSMM: Skeletal Muscle Mass Percentage. The ANOVA test was used for comparing independent and related samples. ^a^ pre vs. post *p* value; ^b^
*p*-value between groups. Statistical significance was considered at a *p* value of <0.05.

**Table 3 jfmk-09-00004-t003:** Intra and intergroup comparison of changes in blood count.

	CG (*n* = 7)			REG (*n* = 7)			CEG (*n* = 9)			*p* Value ^b^
	Pre	Post	*p* Value ^a^	Pre	Post	*p* Value ^a^	Pre	Post	*p* Value ^a^	
Hemoglobin (g/dL)	8.20 (8.9–5.4)	8.60 (6.50–10.0)	0.611	8.00 (6.10–8.70)	7.40 (5.90–9.70)	0.866	6.75 (4.80–12.10)	8.40 (5.70–9.20)	0.018	0.466
Neutrophiles (×10^3^/µL)	2.30 (0.4–89.4)	3.02 (0.10–16.90)	0.116	0.70 (0.4–160.0)	0.20 (0.00–1.50)	0.063	1.20 (0.40–12.10)	0.40 (0.00–6.20)	0.345	0.826
Leucocytes (×10^3^/µL)	46.30 (1.3–386)	8.48 (0.30–47.60)	0.091	19.50 (3.10–282.00)	0.70 (0.20–3.20)	0.028	13.45 (1.80–171.50)	0.90 (0.30–2.50)	0.063	0.263
Platelets (×10^3^/µL)	23.00 (14.0–62.0)	53.00 (18.00–395.00)	0.173	49.00 (8.00–92.00)	44.00 (11.00–365.00)	0.753	35.50 (7.00–109.00)	48.00 (11.00–97.00)	0.236	0.097

The Mann–Whitney U test was used for independent and related samples. ^a^ pre vs. post *p* value; ^b^
*p*-value between groups. Statistical significance was considered at a *p*-value of <0.05.

**Table 4 jfmk-09-00004-t004:** Correlation between exercise intervention and clinical prognosis.

	CG (*n* = 7)	REG (*n* = 7)	CEG (*n* = 9)	*p* Value
Survival n (%)				0.080
Alive	3 (42.14)	6 (85.7)	8 (88.9)	
Death	4 (57.1)	1 (14.3)	1 (11.1)	
Overall survival (days)	412.71 ± 94.33	444.57 ± 146.65	488.55 ± 34.33	0.319
MRD + 45 days n (%)				0.082
Positive	7 (100)	5 (71.4)	9 (100)	
Negative	0 (0)	2 (28.6)	0 (0)	
Relapse n (%)				0.612
Absence	5 (71.4)	5 (71.4)	8 (88.9)	
Presence	2 (28.8)	2 (28.8)	1 (11.1)	
Response to induction n (%)				0.264
Remission	4 (57.1)	6 (85.7)	8 (88.9)	
Refractory	3 (42.9)	1 (14.3)	1(11.1)	

The values are expressed in the number of cases and (%) for the categorical variables and mean and ± SD for the quantitative variables. Chi-square tests were performed to compare the proportions of the intergroup variables, and a three-way ANOVA was used to compare intergroup means. Statistical significance was considered at a *p* value of <0.05.

## Data Availability

Data is available on request due to restrictions of privacy and confidentiality manners.

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
