# Peer review of "Effect of a Cross-Training and Resistance Exercise Routine on IL-15 in Adults with Type B Acute Lymphoblastic Leukemia during the Induction Phase: Randomized Pilot Study"

_jfmk, 2023, doi:10.3390/jfmk9010004_

Round 1
Reviewer 1 Report
Comments and Suggestions for Authors
I have read with great interest the paper submitted by Gallardo Rodríguez et al. The topic is interesting and the authors managed to convey this in the introduction. There are some fundamental issues, however, that need to be addressed:
- Further details are required on how the randomization was conducted. Even though the authors show that there are no significant differences between lots, one can observe that the composition is different in terms of age, gender, BMI, Hb, PLT, Leu, specific lymphocytes and so on...This begs the question on how the p-values between the three groups were performed, especially in Table 1, where non-parametric data and categorical data is involved. Were the variable tested in pairs?
- Did the authors consider that the CG was not properly selected as they have an above-average physical activity, albeit low-grade intensity, and perhaps a CG with no intervention would be better suited for this study? Can the authors report what statistical differences were between the Healthy lot (n=5) and each of the three study lots in terms of IL-15? This needs to be further discussed in the Materials and methods section of in the Discussions, comparing the approach to similar studies.
- Does the "within-groups" testing (Tables 2,3) refer to pre- vs post- testing? If so, please make this more obvious by adding further details into the captions.
- Which statistical tests were used in Table 4 to compare overall survival between the lots?
- The discussions are interesting but may be difficult to navigate by certain readers, especially if not particularly familiar with the topic; I recommend the authors include a Figure depicting the role of IL-15 in the intermediary and energy metabolisms and the interplay with ALL, the nervous system and cellular component of the immune response.
- minor spelling and grammar issues need to be addressed.
Overall, the paper looks promising and I congratulate the authors for their effort.
Comments on the Quality of English LanguageMinor spelling and grammar issues need to be addressed.
Reviewer 2 Report
Comments and Suggestions for Authors
In this paper, Adan and colleagues present a study examining the impact of exercise intervention on IL-15 levels in acute lymphoblastic leukemia patients. The study concludes that resistance exercises might enhance survival prognosis and reduce relapses. However, the author acknowledges limitations, specifically the small patient sample size and considerable variability in IL-15 observations (Fig. 1). Notably, IL-15 production appears to be closely tied to changes in immune components, particularly since the patients underwent chemotherapy during the exercise intervention. Examination of the tables reveals significant changes in leukocyte, platelet, and neutrophil counts due to chemotherapy. The observed variations stemming from chemotherapy across patients make it challenging to draw conclusive findings regarding the effect of exercise on IL-15 in the current setup. Consequently, the reviewer suggests rejecting the manuscript.
Comments on the Quality of English Languagen.a.
Reviewer 3 Report
Comments and Suggestions for Authors
1. What happened to those 10 missing patients? Did they expire during the trial, prior to 2nd blood draw? Or did they resign from participation?
2. Three groups differed quite significantly in terms of NEU and LEU - why?
3. Table 1 - unit is missing for IL-15
4. What standards were used for IL-15? Why didn't the authors use a ready ELISA kit for IL-15?
Professional proof-reading highly suggested
Round 2
Reviewer 1 Report
Comments and Suggestions for Authors
The authors have performed adequate corrections to the manuscript per the suggestions provided.
Reviewer 2 Report
Comments and Suggestions for Authors
The reviewer comprehends the difficulty and acknowledges the authors' explanation.